# Functional crosstalk between the cohesin loader and chromatin remodelers

Sofía Muñoz [1,2] ✉, Andrew Jones[3], Céline Bouchoux [1], Tegan Gilmore [4], Harshil Patel[4] & Frank Uhlmann [1] ✉

The cohesin complex participates in many structural and functional aspects of genome organization. Cohesin recruitment onto chromosomes requires nucleosome-free DNA and the Scc2-Scc4 cohesin loader complex that catalyzes topological cohesin loading. Additionally, the cohesin loader facilitates promoter nucleosome clearance in a yet unknown way, and it recognizes chromatin receptors such as the RSC chromatin remodeler. Here, we explore the cohesin loader-RSC interaction. Amongst multi-pronged contacts by Scc2 and Scc4, we find that Scc4 contacts a conserved patch on the RSC ATPase motor module. The cohesin loader directly stimulates in vitro nucleosome sliding by RSC, providing an explanation how it facilitates promoter nucleosome clearance. Furthermore, we observe cohesin loader interactions with a wide range of chromatin remodelers. Our results provide mechanistic insight into how the cohesin loader recognizes, as well as influences, the chromatin landscape, with implications for our understanding of human developmental disorders including Cornelia de Lange and Coffin-Siris syndromes.

Eukaryotic genomes are packed into chromatin. The basic unit of chromatin is the nucleosome, ~147 bp of DNA wrapped around a histone octamer. Nucleosomes hinder DNA access to enzymes involved in DNA metabolism such as transcription, replication, or DNA repair. Thereby, nucleosome positioning plays a critical role in genome regulation. Nucleosome positions along the genome are determined by several factors: the underlying DNA sequence, transcription factors, histone post-translational modifications and the action of chromatin-remodeling complexes[1,2]. While the relative contribution of each of these factors and their interdependencies are incompletely understood, abundant genetic and biochemical evidence points to the importance of chromatin remodelers, molecular machines that utilize the energy from ATP hydrolysis to translocate along DNA. Diverse protein domains and subunits within different remodeler families flank a conserved ATPase motor domain, giving rise to distinct remodeling outcomes such as nucleosome sliding, nucleosome spacing, histone octamer ejection or histone variant exchange, which together generate a dynamic chromatin landscape.

An additional layer of DNA organization is accomplished by folding the nucleosome fiber through DNA looping and DNA-DNA interactions. This task relies on SMC (Structural Maintenance of Chromosomes) complexes, conserved ring-shaped protein complexes powered by ATP hydrolysis, that are able to encircle DNA[3–5]. Amongst the SMC complexes, cohesin holds together the two sister chromatids following DNA replication in a process known as sister chromatid cohesion, vital for accurate chromosome segregation during cell divisions[3,6,7]. In addition to sister chromatid cohesion, cohesin plays central roles in genome organization by loop formation during interphase, transcriptional regulation, as well as DNA repair by homologous recombination[8,9]. Cohesin loading onto chromosomes requires an additional cohesin loader complex comprised of the Scc2 and Scc4 subunits[10].

In vitro, Scc2-Scc4 loads cohesin onto DNA in a sequence-independent manner[11], whereas in vivo, cohesin is thought to be loaded at specific chromosomal locations such as centromeres and the promoters of highly transcribed genes[12–15]. In the budding yeast *Saccharomyces cerevisiae*, these specificities arise from direct protein

[1]Chromosome Segregation Laboratory, The Francis Crick Institute, London, UK. [2]Cell Cycle Control and the Maintenance of Genomic Stability Laboratory, Cancer Research Center (CIC), University of Salamanca, Salamanca, Spain. [3]Proteomics Science Technology Platform, The Francis Crick Institute, London, UK. [4]Bioinformatics & Biostatistics Science Technology Platform, The Francis Crick Institute, London, UK. ✉e-mail: sofiamf@usal.es; frank.uhlmann@crick.ac.uk

interactions of the Scc4 cohesin loader subunit with chromatin receptors, including the Ctf19 inner kinetochore complex and the RSC (Remodels the Structure of Chromatin) chromatin remodeling complex[13,16]. RSC, on the one hand, acts as a docking platform that recruits the cohesin loader via a direct protein interaction. In addition, chromatin remodeling by RSC is required to generate a nucleosome-free region that is the substrate for cohesin loading[17]. The relationship between RSC and Scc2-Scc4 extends beyond facilitating cohesin loading - the cohesin loader in turn feeds back onto the nucleosome landscape. Depletion of the cohesin loader[13], or of cohesin itself[18], lead to compromised promoter nucleosomes clearance and altered transcriptional programs, suggesting a two-ways functional crosstalk between primary chromatin structure and cohesin function. However, the mechanism by which the cohesin loader and cohesin feed back onto the chromatin landscape remains incompletely understood.

Here we map the multipronged interactions between the cohesin loader and RSC. Amongst the interactions, Scc4 directly contacts a conserved patch on the RSC ATPase module. We find that the cohesin loader directly stimulates RSC nucleosome remodeling in vitro and facilitates promoter nucleosome eviction in vivo. The cohesin loader interaction extends to members of additional chromatin remodeler families, suggesting a more general way in which cohesin accesses chromatin, potentially also outside of promoter regions.

## Results

### Interactions between the cohesin loader and the RSC chromatin remodeler

RSC acts as a chromatin receptor for the cohesin loader via interactions that involve both Scc2 and Scc4 subunits[17]. RSC itself is a multisubunit complex composed of 16 different protein subunits, providing opportunities for multiple protein contacts. To better understand the configuration of these interactions we employed protein crosslinking mass spectrometry (CLMS).

Equimolar amounts of purified budding yeast RSC and Scc2-Scc4 were incubated with the chemical crosslinker disuccinimidyl sulfoxide (DSSO), an amine reactive crosslinker that covalently bonds lysines that lie spatially close within a protein complex (Fig. 1a, Supplementary Fig. 1a). The crosslinked proteins were digested and analyzed by tandem mass spectrometry to identify crosslinked peptides (Supplementary Data 1). Many of the detected crosslinks were among subunits of the RSC complex, consistent with the known RSC subunit arrangement revealed in recent structural studies[19,20]. In addition, we detected crosslinks between surface residues on both the Scc2 and Scc4 subunits of the cohesin loader and Sth1, the large ATPase subunit of the RSC remodeler complex (Fig. 1b).

As a complementary approach to protein crosslink mass spectrometry, we probed the cohesin loader-RSC interaction using peptide arrays. Overlapping 20-amino-acid-long peptides covering the amino acid sequences of the cohesin loader subunits Scc2 and Scc4 were synthesized on cellulose membranes. These peptide arrays were incubated with the purified RSC complex that we subsequently detected by immunoblotting (Supplementary Fig. 1b, c). Conversely, we prepared a peptide array of Sth1 that we probed by incubation with the Scc2-Scc4 cohesin loader (Supplementary Fig. 1d). These approaches revealed a number of potential, surface-exposed interaction sites, illustrated on structural models of RSC and the cohesin loader in Fig. 1c. Notably, interactions identified by CLMS on Scc4, Scc2 and the Sth1 ATPase module were each flanked by sequences identified in the peptide scans.

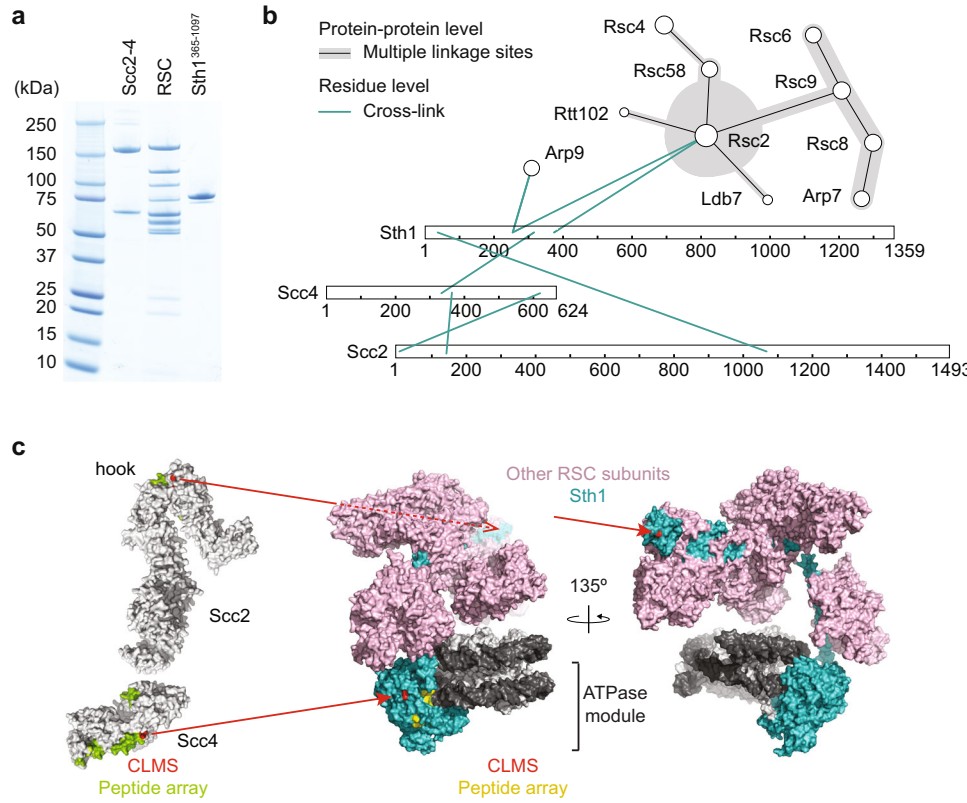

**Fig. 1 | Protein interactions between the RSC chromatin remodeler and the cohesin loader. a** Purified Scc2-Scc4 and RSC complexes, as well as the Sth1[365–1097] fragment were analyzed by SDS-PAGE followed by Coomassie Blue staining. A representative gel from three purifications is shown. **b** Protein contacts, detected by CLMS, within the RSC complex and between RSC and the cohesin loader. Data from two biological repeat experiments was combined. Sth1, Scc4 and Scc2 are shown in extended form. **c** Position of the CLMS interactions from **b**, as well as candidate interactions based on the peptide array analysis shown in Supplementary Fig. 1, mapped onto structural models of the cohesin loader (PDB: 4XDN for Scc4-Scc2N, together with an Scc2C model built using Phyre2[57] based on PDBs: 5ME3 and 5T8V) and RSC (PDB: 6KW3). Source data are provided as a Source Data file.

## RSC interactions with both Scc2 and Scc4

The cohesin loader contains two functional modules. Firstly, a globular head comprised of Scc4, encapsulating the Scc2 N-terminus (Scc4-Scc2N), required for cohesin loader recruitment to its chromatin receptors[21,22]. An Sth1 interaction revealed by both CLMS and peptide array analyses mapped to a region on Scc4 that also interacts with the inner kinetochore protein Ctf19[16] (Fig. 1c). A small deletion in this area of the human Scc4 ortholog, MAU2, causes Cornelia de Lange Syndrome (CdLS)[23]. These observations open the possibility that Scc4 uses a specialized patch to interact with varied chromatin receptors, whose mutation impairs cohesin loader function.

The remaining C-terminal part of Scc2 (Scc2C), consisting of HEAT repeats, interacts with cohesin and DNA to catalyze the cohesin loading reaction[24–26]. In addition, Scc2C also participates in the cohesin loader-RSC interaction[17]. We recorded a CLMS interaction, next to peptide array signals, on a region of Scc2C known as the Scc2 'hook' (Fig. 1c). The Euclidean distance between the Scc2 hook and Scc4 is compatible with the distance between their respective interaction sites on Sth1, the ATPase module and its Sth1 N-terminal extension, respectively. This arrangement suggests that both cohesin loader-RSC interactions can occur simultaneously. When engaging cohesin and DNA, the cohesin loader uses surfaces distinct from the RSC interaction sites[24,26]. This observation opens the possibility that cohesin loading onto DNA occurs while the cohesin loader retains contact with its chromatin receptors.

## The cohesin loader interaction with the RSC ATPase module

The cohesin loader binding sites on the RSC ATPase module, identified by CLMS and peptide arrays, lie in the 'post HSA', 'protrusion' and 'braces' regions of Sth1 (Fig. 2a). These regions spatially colocalize, forming a bundle of α-helices that acts as a structural hub to regulate the remodeler ATPase and DNA translocation efficiency[27,28]. To query whether the cohesin loader indeed binds to the Sth1 motor module, we expressed and purified a recombinant Sth1 fragment (Sth1[365–1097], spanning residues 365-1097) that includes the two RecA-like ATPase lobes and regulatory regions (Figs. 1a, 2a). We then evaluated the ability of the Sth1[365–1097] fragment, in comparison with the RSC holo-complex, to interact with the purified cohesin loader. Immunoprecipitation of the cohesin loader led to co-purification of RSC (Fig. 2b). Cohesin loader pulldown similarly recovered the isolated Sth1[365–1097] fragment, suggesting that a prominent cohesin loader contact within the RSC chromatin remodeling complex lies within the Sth1 ATPase module.

To ask whether the Sth1[365–1097] interaction is exclusive to Scc4, we purified the Scc4-Scc2N and Scc2C cohesin loader modules[22,25]. The intact Scc2-Scc4 cohesin loader and both modules were then immunoprecipitated using magnetic beads, incubated with Sth1[365–1097] and the bound proteins were visualized by Coomassie blue staining. Sth1[365–1097] bound most strongly to the full Scc2-Scc4 complex, while both Scc4-Scc2N and Scc2C modules interacted to a lesser extent (Fig. 2c). Considering our previous identification of an Scc2 hook interaction with the Sth1 N-terminus, these results suggest that Scc2 engages in additional contacts with the Sth1 ATPase. The results from our peptide scanning approach (Supplementary Fig. 1c) highlight candidate regions for such additional interactions.

We next explored whether the ATP-binding state of the Sth1[365–1097] ATPase influences its interaction with the cohesin loader. To this end, we repeated the immunoprecipitation experiment in the absence or presence of ATP, or in the presence of the non-hydrolyzable ATP mimetic ADP· BeF$_3$. The presence of the non-hydrolyzable ATP analogue markedly increased the robustness of the Sth1[365–1097] interaction with the Scc2-Scc4 cohesin loader, as well as with the individual Scc4-Scc2N and Scc2C modules (Fig. 2d). This suggests that the cohesin loader preferentially binds to the ATP-bound state of the RSC ATPase. We noticed that the presence of ADP· BeF$_3$ not only stabilized the Sth1-cohesin loader interaction, but also increased the protein melting temperature of the recombinant Sth1[365–1097] fragment by more than 4 °C (Supplementary Fig. 2). Taken together, these observations

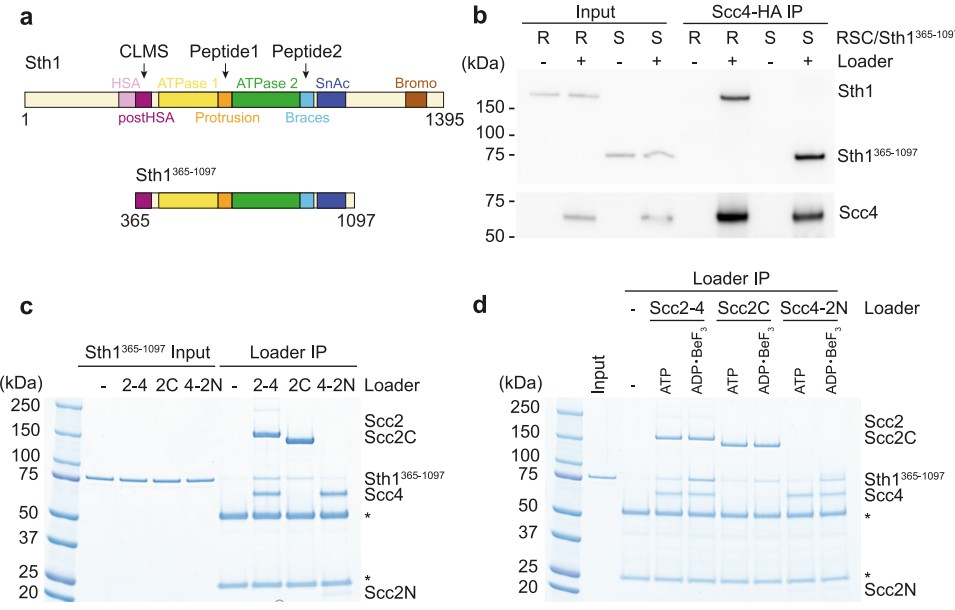

**Fig. 2 | Scc2-Scc4 binds the RSC ATPase domain. a** Schematic representing Sth1 domain architecture and the Sth1[365–1097] construct (CLMS, cross linking mass spectrometry; HSA, helicase-SANT-associated; SnAc, Snf2 ATP coupling domains). The positions of the CLMS contact and peptide array interactions are indicated. **b** Interaction between Scc2-Scc4 and Sth1[365–1097]. Equimolar amounts of cohesin loader and either the RSC complex or Sth1[365–1097] were mixed, followed by cohesin loader immunoprecipitation (IP). Sth1 coprecipitation was analyzed by immunoblotting. **c** Sth1[365–1097] interaction with Scc2-Scc4 (2-4), Scc2C (2C) or Scc4-Scc2N (4-2 N), coupled to anti-HA antibody-coated magnetic beads. Recovered protein was visualized by Coomassie Blue staining. * Asterisks indicate heavy and light chains of the antibody used for immunoprecipitation. **d** Preferential interaction in a nucleotide-bound state. Immunoprecipitation experiments in **c** were repeated in the presence of either ATP or ADP· BeF$_3$. The experiments shown in panels **b**–**d** were twice repeated with similar results. Source data are provided as a Source Data file.

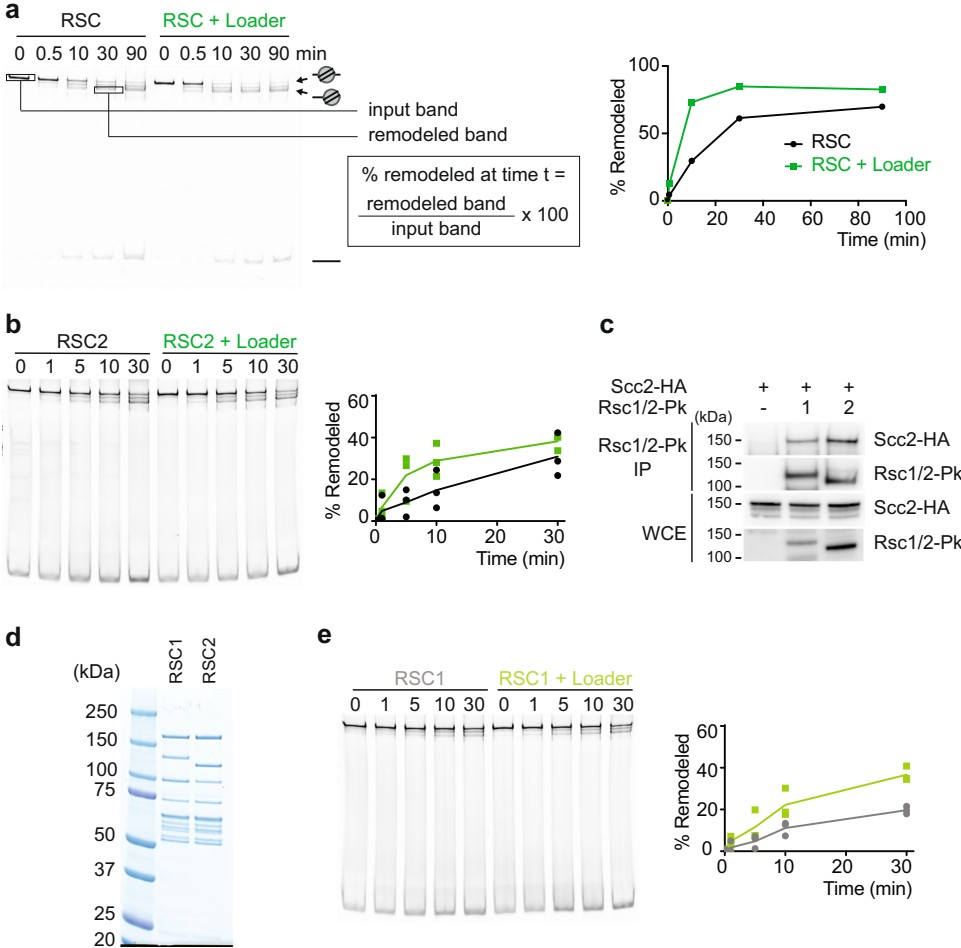

**Fig. 3 | The cohesin loader stimulates RSC nucleosome remodeling. a** RSC-dependent nucleosome sliding assay. Native polyacrylamide gel analysis of the nucleosome substrate and remodeled product (highlighted by arrows). The intensity of the indicated bands was quantified over time. **b** Three additional biological repeats of the assay shown in **a** were performed. A representative gel image, as well as the quantification of each repeat are shown. A line connects the means ($p = 0.0068$, two-way ANOVA test). **c** RSC1 and RSC2 interact with the cohesin loader. Levels of both the Rsc1 and Rsc2 subunits in whole cell extracts (WCE), and co-immunoprecipitation (IP) of the RSC1 and RSC2 complexes with the cohesin loader (detected by an HA epitope on its Scc2 subunit) were analyzed by immunoblotting. The experiment was repeated with similar results. Additionally, RSC1 and RSC2 interaction with the cohesin loader was confirmed using an HA epitope on its Scc4 subunit (Supplementary Fig. 4b). **d** Purified RSC1 and RSC2 complexes, visualized by SDS-PAGE and Coomassie Blue staining. A representative gel from three purifications is shown. **e** RSC1-stimulated nucleosome sliding in the presence or absence of the cohesin loader. Nucleosome substrate and remodeled products during a representative timecourse experiment are shown. The graph presents the results from three biological repeat experiments. A line connects the means ($p = 0.0048$, two-way ANOVA test). Source data are provided as a Source Data file.

suggest that the Sth1 ATPase is stabilized in the presence of ADP · BeF$_3$ in a state that is recognized by the Scc2-Scc4 cohesin loader.

## The cohesin loader stimulates RSC chromatin remodeling activity

Since the cohesin loader physically contacts the RSC ATPase domain, we investigated whether the cohesin loader influences RSC chromatin remodeling activity. Using a synthetic mononucleosome substrate, the ability of RSC to promote nucleosome sliding can be monitored[29]. We prepared such a substrate by mixing recombinant yeast histone octamers with a DNA fragment containing a centrally located Widom 601 nucleosome positioning sequence and salt gradient dialysis[30,31]. Following incubation with RSC and ATP, nucleosome remodeling can be visualized by native polyacrylamide gel electrophoresis, where repositioned nucleosomes migrate faster than the centrally positioned starting nucleosome (Fig. 3a)[32]. To a lesser extent, nucleosome eviction also becomes apparent in these assays, resulting in release of free DNA. As nucleosome eviction remained unaffected by the cohesin loader, we will in the following focus our analysis on nucleosome repositioning. Consistent with the expected properties of the remodeling

reaction, changes to the nucleosome substrate occurred only in the presence of both RSC and of ATP in the incubation (Supplementary Fig. 3a, b).

To test whether Scc2-Scc4 has an effect on RSC chromatin remodeling, we repeated the above assay in the presence or absence of the cohesin loader (10 nM RSC is used in the remodeling assay[32], which we supplemented with 15 nM of the Scc2-Scc4 complex). Cohesin loader addition reproducibly led to the accelerated appearance of remodeled products (Fig. 3a, three biological repeats and their quantification are presented in Fig. 3b). These observations revealed that the cohesin loader directly stimulates nucleosome remodeling by RSC.

To understand whether RSC stimulation is confined to one of the two cohesin loader modules, we repeated the nucleosome sliding assays in the presence of either Scc2C or Scc4-Scc2N. Addition of either Scc2C or Scc4-Scc2N somewhat enhanced nucleosome sliding, albeit to a lesser extent than addition of the full Scc2-Scc4 complex (Supplementary Fig. 3c). This observation suggests that the two cohesin loader modules play a joint role in stimulating RSC nucleosome remodeling activity.

Despite boosting RSC remodeling activity, neither the full Scc2-Scc4 complex, nor either of the cohesin loader modules, significantly changed the RSC ATP hydrolysis rate (Supplementary Fig. 4a). This result indicates that the cohesin loader stimulates RSC remodeling not by increasing ATP hydrolysis, but rather by improving the coupling between ATP hydrolysis and DNA translocation, a common mode of RSC regulation[32]. In other words, Scc2-Scc4 does not change the ATP hydrolysis rate, but enhances the efficiency of DNA translocation during each ATP hydrolysis cycle.

## Both RSC1 and RSC2 complexes are stimulated by the cohesin loader

Two distinct RSC complexes exist in budding yeast, characterized by the presence of one of the two paralogous Rsc1 or Rsc2 subunits. The RSC2 complex is most often studied, containing the Rsc2 subunit, which we used for our biochemical experiments until here. The RSC1 complex, carrying the Rsc1 subunit, is less abundant in cells than RSC2 (Fig. 3c). RSC1 is specialized in remodeling partially unwrapped nucleosomes, such as nucleosomes positioned on stiff (AT-rich) DNA sequences[33]. Because the cohesin loader is enriched at polyA tracts[13], we asked whether it preferentially interacts with or activates RSC1.

We first fused Pk epitope tags to either Rsc1 or Rsc2 and performed co-immunoprecipitation experiments with the cohesin loader. Pulldown of either Rsc1 or Rsc2 resulted in comparable co-precipitation of the cohesin loader subunit Scc2 (Fig. 3c). When we repeated Rsc1 and Rsc2 pulldown and probed for co-precipitation of Scc4, we obtained the same result (Supplementary Fig. 4b). This observation suggests that the cohesin loader interacts with both RSC isoforms. Our finding that the cohesin loader contacts the Sth1 subunit, which is common to both RSC1 and RSC2, is consistent with this conclusion.

We next purified the RSC1 complex, after fusing Rsc1 to a tandem affinity purification (TAP) tag, analogous to the established purification protocol for RSC2[29] (Fig. 3d). In agreement with previous observations[33], RSC1 remodeled the nucleosome positioned on the Widom 601 sequence in a slightly less efficient manner when compared to RSC2. Addition of the cohesin loader substantially enhanced the RSC1 sliding rate, such that it approached the rate observed with RSC2 (Fig. 3e, Supplementary Fig. 4c). Together, these results suggest that the cohesin loader does not discriminate between the two RSC isoforms and stimulates chromatin remodeling by both the RSC1 and RSC2 complexes.

## Both cohesin loader modules contribute to in vivo nucleosome positioning

A yeast strain carrying a thermosensitive cohesin loader mutation, scc2-4, displays increased nucleosome occupancy at gene promoters at the restrictive temperature, as well as gene expression changes similar to those resulting from RSC inactivation[13]. To study the contribution of the two cohesin loader modules to in vivo nucleosome positioning, we utilized conditional Scc2 depletion from yeast cells using promoter shut-off coupled to auxin-mediated Scc2 degradation[17]. First, we confirmed that Scc2 depletion results in a similar effect on nucleosome positioning as was seen with the scc2-4 allele. 30 min following promoter repression and auxin addition to G1 arrested cells, Scc2 reached close to the background levels (Supplementary Fig. 5). Micrococcal nuclease digestion followed by high throughput sequencing were now used to generate nucleosome maps. Confirming previous observations, cohesin loader depletion led to increased nucleosome occupancy at gene promoters. This became apparent from heatmaps of all gene promoters (Supplementary Fig. 6a), or when plotting average nucleosome occupancy surrounding the most affected gene promoters, or in a profile overlay of all those promoters previously identified as cohesin loader binding sites[13] (Supplementary Fig. 6b).

We then addressed which of the cohesin loader modules, Scc4-Scc2N or Scc2C contribute to in vivo nucleosome remodeling. To do so, we ectopically expressed either a wild-type copy of Scc2, or one of its two functional units Scc2N or Scc2C, in the Scc2 depletion strain background (expression of Scc2N recreates the Scc4-Scc2N module in Scc2-depleted cells)[17]. We then assessed the ability of the two cohesin loader modules to rescue the nucleosome positioning defect caused by Scc2 depletion. This analysis revealed that expression of either Scc2N or Scc2C accomplished only a partial restoration of the nucleosome-depleted region at gene promoters (Fig. 4a, b). In contrast, expression of full-length Scc2 resulted in close to complete rescue of the wild type nucleosome landscape. These observations strengthen the conclusion that both functional units of the cohesin loader, Scc4-Scc2N and Scc2C, act together to promote RSC function in nucleosome sliding to maintain nucleosome-depleted gene promotes.

## Additional chromatin remodelers as cohesin loader receptors

Chromatin remodelers are classified within four families: imitation switch (ISWI), chromodomain helicase DNA-binding (CHD), switch/sucrose non-fermentable (SWI/SNF), and INO80, on the basis of the domain architecture of their ATPase subunits (Fig. 5a). Irrespective of the family, all remodelers are based on a conserved SNF2-related ATPase domain. An ISWI family remodeler has previously been implicated in loading of the human cohesin complex onto chromosomes[34], while the budding yeast ISW1 and CHD1 remodelers can replace RSC as receptors for the cohesin loader[17]. Our finding that the cohesin loader engages RSC via its conserved ATPase domain opened the possibility that the cohesin loader might interact with additional chromatin remodelers.

To test whether chromatin remodelers other than RSC interact with the cohesin loader, we created a series of budding yeast strains in which one of each of the chromatin remodeler ATPases was fused to a protein A tag. Affinity pulldown of the different chromatin remodelers revealed that the cohesin loader, detected by its Scc4 subunit, was recovered in variable efficiencies with each of them, but not in a control pulldown from a strain lacking a protein A tag (Fig. 5b). Benzonase was included in all steps of extract preparation and pulldowns, to obviate indirect, chromatin-mediated interactions in this assay. Thus, the cohesin loader interacts more broadly with a range of chromatin remodelers.

To investigate whether interactions between cohesin loader and chromatin remodelers are conserved in other species, we likewise tagged all remodeler ATPase subunits in the fission yeast S. pombe with a protein A tag and assessed cohesin loader co-precipitation. This confirmed that the fission yeast Scc4 ortholog, Ssl3, physically interacts with the fission yeast RSC complex[35], and revealed that Ssl3 also interacts with remodelers from other families (Fig. 5c). Thus, an interaction between the cohesin loader and chromatin remodelers appears conserved not only amongst remodeler families but also amongst species.

## Conserved sequence motifs underpin the Scc4 – RSC interaction

To survey commonalities amongst chromatin remodelers that might explain a common cohesin loader interaction, we studied the sequence conservation around the cohesin loader interaction site on the Sth1 ATPase, identified in our CLMS and peptide array experiments. This analysis revealed a highly conserved FEDWF motif within the region identified by the peptide scan, both amongst chromatin remodeler families in S. cerevisiae and between species (Fig. 5d). The motif is part of the Sth1 protrusion domain and includes a surface-exposed aromatic tryptophan, as well as negatively charged residues (Supplementary Fig. 7a). To explore the contribution of the FEDWF motif to the cohesin loader interaction, we synthesized an array of mutational peptides in which each of the 20 amino acid positions in

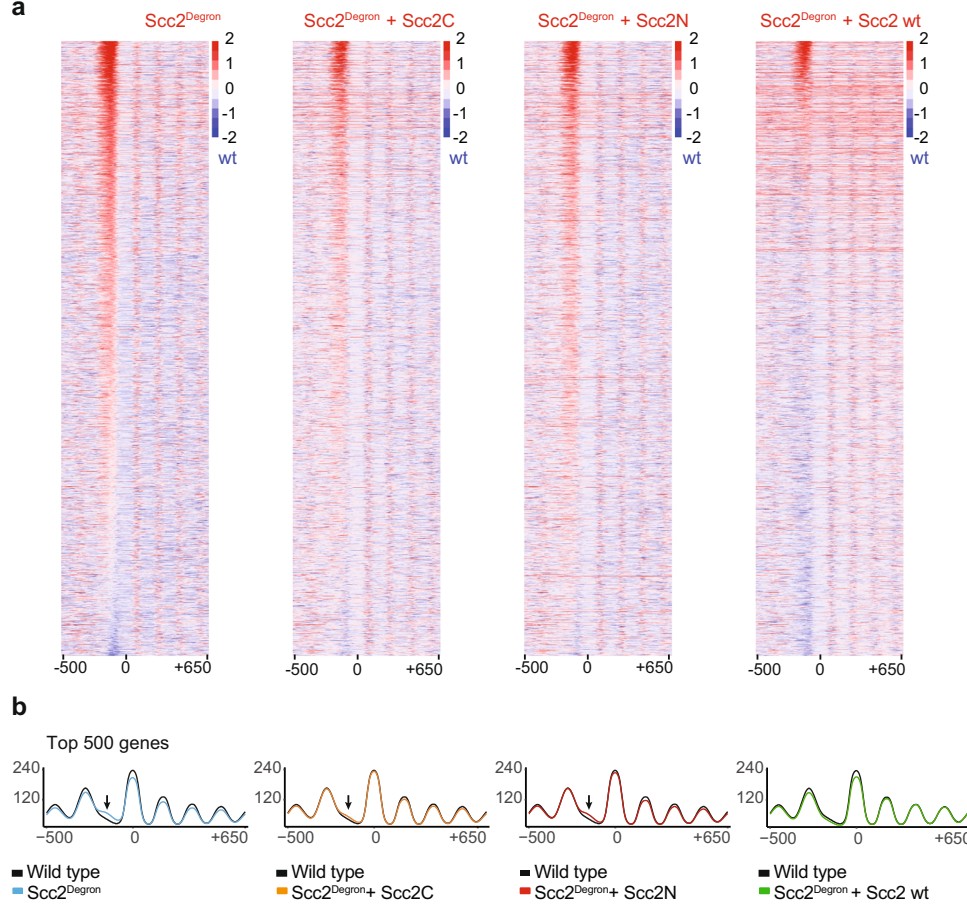

**Fig. 4 | Promoter nucleosome eviction by the cohesin loader. a** Differential nucleosome positioning heat maps, centered on the +1 nucleosome midpoint, comparing the indicated conditions to a wt control strain (scale, log$_2$). **b** Average nucleosome occupancy profiles at the 500 gene promoters that showed the greatest difference (within a −150 to −50 interval) between degron and wild type.

The arrows point towards increased nucleosome occupancy. Average nucleosome profiles at all genes, and at all those genes previously identified as cohesin loader binding sites[13] can be found in Supplementary Fig. 6c. A repeat experiment is shown in Supplementary Fig. 6a, b.

the initially identified peptide were mutated to five alternative residues (leucine, glycine, arginine, glutamate or tyrosine). This analysis strikingly confirmed the importance of the FEDWF motif, as well as three succeeding amino acids, for the cohesin loader interaction (Supplementary Fig. 7b). Only a negatively charged glutamate could replace amino acids at most positions, while tyrosine could replace either of the phenylalanines. This points to an important contribution of negative surface charge to the cohesin loader interaction.

Finally, we also inspected the Scc4 surface involved in the Sth1 interaction. While Scc4 is poorly conserved at the primary amino acid level[36], weak similarities could be detected surrounding the RSC interaction site identified by CLMS and the peptide array. We again used mutational peptide arrays to interrogate sequence requirements for the Scc4-RSC interaction. This revealed neighboring SRK and KLIK motifs in Scc4 as determinants of the interaction. Of these, the SRK motif overlaps the known Scc4 interaction site with its Ctf19 centromere receptor[16]. The KLIK motif in turn is conserved amongst species and includes the lysine residue identified in the CLMS interaction. In summary, the cohesin loader interaction with RSC involves specific peptide sequences, the conservation of which is consistent with widespread interactions between the cohesin loader and chromatin remodelers.

## Discussion

The RSC chromatin remodeling complex acts as a chromatin receptor for cohesin loading, both by physically recruiting the cohesin loader, as well as by generating a nucleosome-free DNA template. Here, we

pinpoint a reciprocal function, the cohesin loader stimulates the nucleosome sliding activity of the RSC complex. This reveals a previously unknown biochemical activity contained in the cohesin loader, with implications for the interplay between cohesin loading and gene regulation.

### Allosteric regulation of a chromatin remodeler

The mechanism of how chromatin remodelers act as molecular machines that utilize the energy from ATP hydrolysis to mobilize nucleosomes has been outlined[2,37]. Less is yet known about the regulation of chromatin remodeler action, due to their structural complexity and the numerous combinatorial influences, including DNA sequence context, histone variants and modifications. The structures of several remodeler complexes[38–41] provide a foundation for understanding how these enzymes are regulated, how their different subunits and domains integrate external input and how they finetune the coupling of DNA translocation to desired outcomes in the chromatin landscape. In how far proteins additional to the above chromatin factors contribute to regulating chromatin remodelers is incompletely understood.

Here we show that the budding yeast Scc2-Scc4 cohesin loader stimulates RSC chromatin remodeling activity in vitro by directly engaging with a conserved regulatory hub on the RSC ATPase subunit. The cohesin loader stimulates remodeling without increasing the RSC ATP hydrolysis rate, suggesting that it acts by increasing the coupling of ATP hydrolysis with DNA translocation. Similar behavior was found

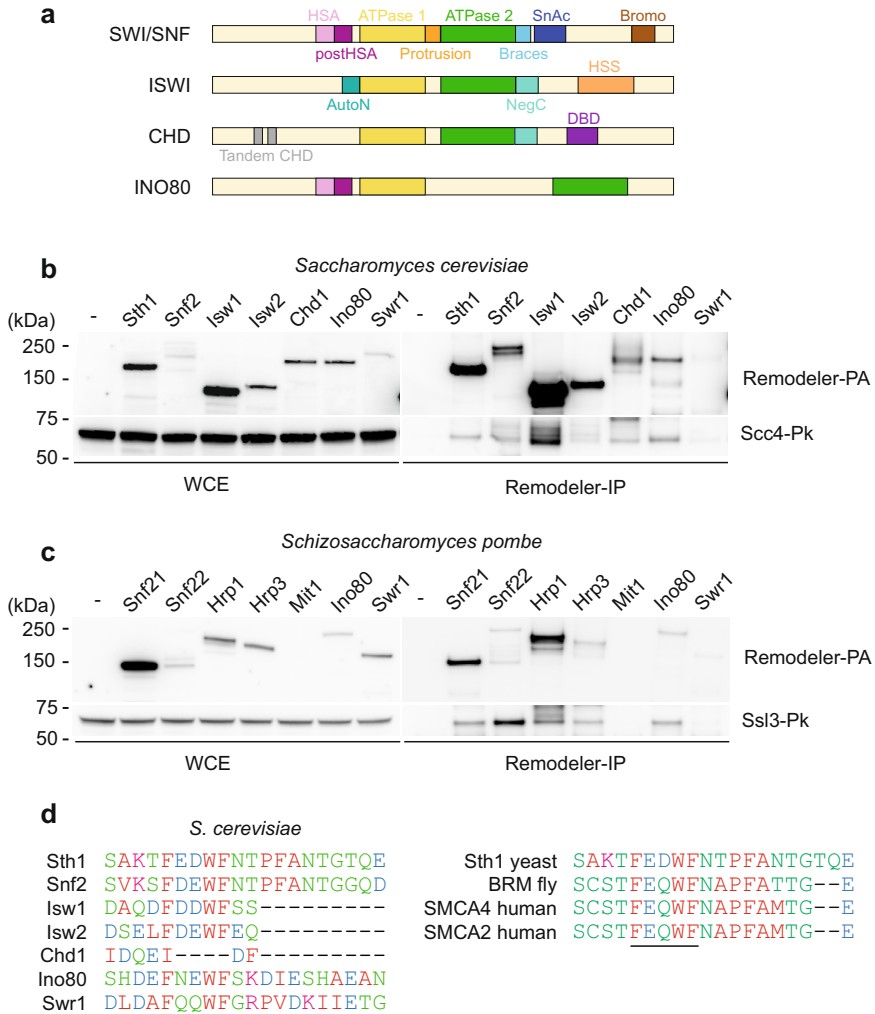

**Fig. 5 | Widespread cohesin loader interactions with chromatin remodelers.**
**a** Domain organization of the ATPase subunits of the four remodeler families.
**b** Interactions between the *S. cerevisiae* remodeler ATPase subunits, pulled down via protein A tags, and the cohesin loader, detected by immunoblotting against Scc4. **c** As in **b**, but precipitating the ATPase subunits of *S. pombe* chromatin remodelers and assessing coprecipitation of the Scc4 ortholog Sls3. These immunoprecipitation (IP) experiments were performed once. **d** Conservation of an FEDWF motif (underlined) amongst remodeler families and organisms. Clustal (vOmega) was used to prepare the alignment. Source data are provided as a Source Data file.

in the cases of several gain-of-function mutations within this regulatory hub, some of which are seen replicated in chromatin remodeler mutations associated with malignant tumors[28,32]. The delineated cohesin loader interaction on the Sth1 ATPase includes a conserved FEDWF motif. Amino acids within this motif are known to play critical roles in RSC regulation, in particular the surface-exposed tryptophan that is essential for in vivo RSC function[28]. Our findings open the possibility that external protein factors regulate RSC in an allosteric manner by contacting this regulatory hub. In case of the cohesin loader, the stimulation of nucleosome sliding by RSC will aid the generation of a stretch of nucleosome-free DNA that in turn facilitates cohesin loading.

**Chromatin remodelers as broad targets of the cohesin loader**
In addition to RSC, we found that the cohesin loader engages in physical interactions with members of other chromatin remodeler families. While the various abundances of the different remodelers in our in vivo cohesin loader interaction assay make it hard to compare relative affinities, the ISWI complex emerged as an abundant and efficient Scc2-Scc4 interactor. Synthetic coupling of the cohesin loader to remodelers of several different families has demonstrated their ability to serve as functional cohesin loader receptors, again with the ISWI complex amongst the proficient targets[17]. The conservation of the SNF2-type ATPase domain amongst chromatin remodeler families, including their FEDWF surface motif, suggest that the cohesin loader might contact them in a similar manner as observed in the case of RSC. Even though remodeling outcomes diverge between remodelers, sliding and ejecting nucleosomes is a common theme amongst remodelers, which could be harnessed by the cohesin loader to access DNA for loading the cohesin complex onto DNA. Whether the cohesin loader stimulates the activities of these additional chromatin remodelers, to facilitate the cohesin loading reaction, remains to be explored.

Individual depletion of chromatin remodelers from either budding or fission yeast has revealed defects in sister chromatid cohesion only following removal of the RSC complex, but not any of the other chromatin remodelers[13,35]. However, co-depletion of the second SWI/SNF ATPase Snf2 in budding yeast, together with RSC, resulted in an augmented cohesion defect. This observation is consistent with the possibility that the Snf2 complex participates in cohesin loading, while its contribution is usually overshadowed by RSC. In human cells, the ISWI remodeler promotes cohesin loading onto chromatin[34], suggestive of a putative involvement of this remodeler also in budding

yeast. The respective contributions of a wider range of chromatin remodelers to cohesin loading, individually or in combination, remains to be further investigated. The joint action of chromatin remodelers from more than one family opens the possibility that cohesin loading takes place not only at gene promoters, marked by the RSC complex, but possibly more broadly along active gene bodies where additional chromatin remodelers act[42].

## The cohesin loader and gene regulation

Mutations in MAU2 and NIPBL, the human cohesin loader subunits, are the cause of Cornelia de Lange Syndrome (CdLS), a hereditary disorder whose clinical features are thought to be the consequence of subtle gene expression changes in numerous developmental genes[23,43]. CdLS results in clinical features similar to those observed in Coffin-Siris syndrome, which in turn results from mutations in subunits of the human SWI/SNF chromatin-remodeler family[44–46]. The similar clinical presentation of cohesin loader and chromatin remodeler mutations highlights the functional overlap between the two complexes. How CdLS and Coffin Siris mutations impact on transcription is still incompletely understood. By impeding cohesin loading, both types of mutations might alter transcription by attenuating the creation of cohesin-dependent DNA loops[9,12]. Consistent with this scenario, CdLS mutations have also been found in subunits of the cohesin complex[47].

On the other hand, NIPBL is also known to act as a transcriptional co-regulator in a cohesin-independent manner, in both *Drosophila* and human cells[15,48]. In one example, the cohesin loader was shown to facilitate transcription elongation by releasing paused RNA polymerase II through interaction with the super elongation complex[49]. Our findings that Scc2-Scc4 complex physically interacts with the RSC ATPase module and enhances its remodeling activity point to the possibility that the cohesin loader controls transcription also through the regulation of the nucleosome landscape. If the cohesin loader modulates chromatin remodelers of more than one family, this could contribute to explaining the multitude of small transcriptional changes observed in CdLS patient cells. Further research will be directed at investigating these, not mutually exclusive, means by which the cohesin loader might impact on gene regulation.

# Methods
## Protein purification
The cohesin loader and Scc2C were purified as described[25]. Scc2 and Scc4, or only Scc2C, were overexpressed under control of galactose-inducible promoters in budding yeast. Cells were disrupted in a cryogenic grinder under liquid nitrogen, the frozen cell powder was thawed, and the lysate was clarified by ultracentrifugation. The proteins were purified by sequential protein A affinity adsorption on IgG-agarose (Sigma) followed by 3 C protease elution, HiTrap Heparin HP and finally Superdex 200 Increase 10/300 GL (Cytiva) chromatography on an Äkta purifier controlled by Unicorn (v.7.6.0.1306) software.

Scc4-Scc2N was purified as described[22]. *E. coli* strain Rosetta 2(DE3) pLysS (Millipore) was transformed with a polycistronic vector carrying Scc2N and Scc4 sequences containing an N-terminal 6-His tag followed by a TEV protease cleavage site. Protein expression was induced with 0.4 mM IPTG at an $OD_{600}$ of ~0.5 and cultures were further incubated overnight at 18 °C. Cells were harvested and lysed by sonication. 6-His-tagged Scc4-Scc2N complexes were isolated from the supernatant by binding to TALON Metal Affinity Resin (Clontech), eluted by TEV protease cleavage and further purified by ion exchange chromatography using a HiTrap SP HP column (Cytiva) and size exclusion chromatography on a 16/600 Superdex 200 column.

RSC was purified from budding yeast cells expressing endogenously tandem affinity purification (TAP)-tagged Rsc1 or Rsc2 subunits, as described[29]. Cells were grown in YPD to stationary phase and disrupted in a cryogenic grinder under liquid nitrogen. The frozen cell powder was thawed, and the lysate was clarified by ultracentrifugation.

The complex was purified by binding to IgG-agarose, followed by TEV protease elution, then bound to calmodulin beads (Clontech) in the presence of calcium, eluted in the presence of EGTA and then dialyzed.

The Sth1[361–1097] fragment was cloned into pGEX-6P-2 and expressed in BL21 Star (DE3) pLysS (ThermoFisher) *E. coli* cells. Cells were grown in TB until reaching an $OD_{600}$ of ~0.8 and were induced with 0.1 mM Isopropyl β-D-1-thiogalactopyranoside (IPTG) at 20 °C, grown overnight and broken by sonication. The GST-fusion protein was adsorbed onto a glutathione-sepharose resin (Cytiva), recovered by cleavage with 3 C protease and purified by size exclusion chromatography using a Superdex 200 column.

Yeast histones were expressed in *E. coli* BL21(DE3) RIL co-expressing pCDFduet.H2A-H2B and pETduet.H3-H4 after 4 h IPTG induction at 37 °C as described in[50]. The cells were broken by sonication and the extract clarified by ultracentrifugation. Histone octamers were then purified by sequential chromatography on HiTrap Heparin and Superdex 200 Increase 16/600 GL columns.

Plasmids used for protein purification are listed in Supplementary Table 1.

## Protein crosslink mass spectrometry
0.5 μM cohesin loader and 0.5 μM RSC were mixed in CLMS buffer (22.5 mM HEPES-KOH pH 7.5, 75 mM NaCl, 100 mM potassium acetate, 10% glycerol, 0.5 mM TCEP) for 15 min at 25 °C and then placed on ice for 15 min. 1 mM disuccinimidyl sulfoxide (DSSO) was added and samples were incubated for 15 min at 37 °C, prior to quenching with 0.10% (v/v) hydroxylamine for 15 min at 37 °C. The crosslinked sample was dried using vacuum centrifugation, resuspended in 8 M urea to a concentration of 1 mg/ml, reduced with 2.5 mM TCEP, and alkylated with 5 mM iodoacetamide. The sample was diluted with 50 mM ammonium bicarbonate to reduce the urea concentration to 1 M, then trypsin was added at a 1:50 enzyme:substrate ratio and left to digest overnight at 37 °C. The solution was then acidified, and peptides were purified using $C_{18}$ solid-phase elution tips (Empore, 3 M). Once activated using acetonitrile, the peptides were added to the $C_{18}$ plug and sequentially eluted using 20 μl 5 mM ammonium formate (pH 10) with 8%, 15%, 20%, 40 and 80% (v/v) acetonitrile. The fractions were dried using vacuum centrifugation. Two biological repeats of this experiment were performed.

The fractions were analysed using a Thermo Fisher Orbitrap Lumos Tribrid Mass Spectrometry coupled to an Ultimate 3000 RSLCnano with an EASY-Spray column (2 μm particles, PepMap C18, 100-Å pore size, 50 cm × 75 μm ID) (Thermo Scientific). A flow rate of 0.25 μl/min was used, starting at 98% mobile A (0.1 % formic acid, 5% DMSO, 95% $H_2O$) and 2% mobile B (0.1% formic acid, 5% DMSO, 80% acetonitrile, 10% $H_2O$). Over 80 min, Mobile B was increased to 40% followed by a further increase to 90% over 10 min. Spectra were acquired with a 375 to 1500 m/z (mass-to-charge ratio) acquisition window. The top ten most intense ions with a charge state of +3 or greater were then selected for tandem MS (MS/MS) using data-dependent acquisition and CID (collision-induced dissociation) fragmentation (at 25% normalized collision energy). A mass-shift triggered MS3 HCD (at 42% normalized collision energy) event was subsequently performed on crosslinked peptides. MaxQuant software (v.2.1.4.0) was used for data collection.

For data analysis, Xcalibur raw files were converted into the MGF format using Proteome Discoverer (v2.2, ThermoScientific) and used directly as input files for XLinkX (v2.2). Searches were performed against an ad hoc protein database containing the sequences of the proteins in the complex and a set of randomized decoy sequences generated by the software. The following parameters were used for the searches: Crosslinker: disuccinimidyl sulfoxide (DSSO, + 158.00376 Da, reactivity toward K, S, Y, T and protein N-terminus); crosslinker fragments: alkene (+54.01056 Da), unsaturated thiol (+85.98264 Da), sulfenic acid (+103.9932 Da); crosslink doublets: alkene/unsaturated thiol (mass difference 31.96704 Da) or alkene/sulfenic acid (mass difference

49.98264 Da); MS[1] mass accuracy: 5 ppm; MS[2] mass accuracy: 10 ppm; MS[3] accuracy: 0.5 Da; enzyme: trypsin; maximum missed cleavages: 3; minimum peptide length: 5 amino acids; maximum number of modifications: 4; fixed modifications: carbamidomethylation of cysteine (mass shift +57.021 Da); variable modifications: Methionine oxidation (mass shift +15.995 Da). For the database search, the FDR was set to 1%. To reduce the number of false-positives, crosslinks identified using XLinkX were filtered for an identification score ≥20. Finally, each fragmentation spectrum was manually inspected and validated. The crosslinks were then visualized using xiVIEW[51]. The results from both biological repeats are reported separately and were combined for the presentation in Fig. 1b.

## Peptide arrays

20 amino acid long peptides covering the amino acid sequences of Scc2, Scc4 or Sth1, at 2 amino acids intervals, were synthesized on cellulose membranes using an Intavis Multipep peptide synthesizer (Intavis Bioanalytical Instruments AG). The membrane was activated in 50% methanol/10% acetic acid, then blocked with 2.5% dried milk in TBS (25 mM Tris/HCl pH 7.5, 150 mM NaCl, 0.1% Tween 20) at room temperature for 2 h. The blocked membrane was incubated with 1 µg/ml of either RSC or cohesin loader in 2.5% milk in TBS at 4 °C overnight. The membrane was washed with TBS and either bound RSC was detected using an anti-Pk epitope antibody (clone SV5-Pk1, Biorad MCA1360, used 1: 2000) or the cohesin loader was detected using an anti-HA epitope antibody (clone F7, Santa Cruz Biotechnology sc-7392, used 1: 5,000).

## Interaction analyses using purified proteins

For interaction analyses followed by immunoblotting, 50 nM Scc2-Scc4 and 50 nM RSC (or Sth1[365–1097]) were mixed in 50 µl of IP buffer (25 mM Tris-HCl pH 7.5, 0.5 mM TCEP, 100 mM NaCl, 2.5 mM MgCl₂, 0.2% Triton X-100, 5% glycerol) and incubated at 25 °C for 15 min. After placing on ice for 15 min, the mixtures were transferred to F7 anti-HA antibody-coated, protein A-conjugated magnetic beads (Protein A Dynabeads™, Invitrogene), and rocked 2 h at 4 °C. The beads were washed three times with IP buffer and once with IP buffer containing 300 mM NaCl. The bound proteins were eluted in SDS-polyacrylamide gel electrophoresis (SDS-PAGE) loading buffer.

For analyses using Coomassie Blue staining, 10 pmol of either Scc2-Scc4, Scc2C or Scc4-Scc2N were incubated with F7 anti-HA antibody and protein A-conjugated magnetic beads at 4 °C for 1 h in IP buffer. After three washes with IP buffer, the loader-bound beads were incubated with 5 µM Sth1[365–1097] at 4 °C for 30 min and washed three times with IP buffer before bound proteins were eluted with SDS-PAGE loading buffer.

## Protein stability analysis

A Prometheus NT.48 (NanoTemper) was used to run Tryptophan fluorescence measurements. Standard-grade glass capillaries were filled with 10–15 µl of the sample, excitation light was preadjusted to yield fluorescence readings above 5000 arbitrary units for F330 and F350, and samples were exposed to a temperature gradient from 20–95 °C with a temperature slope of 1.5 °C/min. Sth1[365–1097] thermal unfolding curves were measured in the presence or absence of 5 mM ATP-analog ADP· BeF₃ and their first derivatives displayed.

## Mononucleosome substrate assembly

a 200 bp DNA fragment containing a centrally located Widom 601 nucleosome positioning sequence was produced by PCR amplification using plasmid 1380 (a gift from the Cherepanov lab) as a template. 15.6 µg DNA (120 pmol) were mixed with 15.8 µg histone octamers (144 pmol, 1:1.2 DNA: octamer ratio) in a 150 µl reaction volume in 2 M NaCl, 10 mM Tris HCl pH 7.5, 0.5 mM TCEP and subjected to a linear salt gradient dialysis from 2 M to 50 mM NaCl at 4 °C using Slide-A-Lyser

Mini Dialysis units (ThermoFisher Scientific) with a 3500 molecular weight cut-off. Mononucleosomes were incubated for a final 30 min at 37 °C to position the histone octamer on the Widow 601 sequence.

## Nucleosome sliding assay

Nucleosome sliding assays were performed as described in[32], using a 50 µl starting reaction containing 20 nM mononucleosomal substrate and 10 nM RSC, with or without 15 nM cohesin loader, Scc2C, or Scc4-Scc2N, and incubated in 10 mM HEPES/NaOH pH 7.5, 20 mM KOAc, 45 mM NaCl, 1 mM MgCl₂, 0.1 mg/ml BSA, 1 mM ATP at 30 °C with shaking at 500 rpm in a Thermomixer (Eppendorf). 10 µl aliquots were retrieved at each time point and reactions were terminated by adding 200 ng competitor DNA (pBluescript plasmid). 10% glycerol was added, and samples were loaded onto Novex TBE, 10% native polyacrylamide gels (Invitrogen), subsequently stained with SYBR Gold (Invitrogen) and scanned on a Typhoon FLA 9500 imager using its control software (v.1.1, Amersham, GE). Quantification was performed with ImageQuant TL (v8.1.0.0) software, statistical analyses and representation were conducted with GraphPad Prism (v7.0c).

## ATPase assay

10 nM RSC and 15 nM cohesin loader, Scc2C or Scc4-Scc2N were mixed with 20 nM mononucleosomal substrate in nucleosome sliding buffer. Reactions were initiated by addition of 1 mM ATP, spiked with [γ-³³P]-ATP (Hartmann Analytic), and incubated at 30 °C. Reaction aliquots were taken at 0, 15, 30 and 60 min and stopped by addition of 3 volumes of 500 mM EDTA. 0.5 µl of the terminated reactions were spotted on polyethylenimine cellulose F sheets (Merck) and separated by thin layer chromatography using 400 mM LiCl in 1 M formic acid as the mobile phase. The separated spots representing ATP and released inorganic phosphate were quantified using a Typhoon FLA 9500 imager and ImageQuant TL (v8.1.0.0) software.

## Yeast strains and culture

All *Saccharomyces cerevisiae* yeast strains used in this study were of the W303 background and are listed in Supplementary Table 2. Cells were cultured in rich YP medium or in complete synthetic medium (CSM) lacking methionine, supplemented with 2% glucose at 25 °C. α-factor was used at a concentration of 7.5 µg/ml, and indole-3-acetic acid (IAA) acid at 88 µg/ml. *Schizosaccharomyces pombe* yeast strains used in this study are also listed in Supplementary Table 2. and were grown at 30 °C in YES medium.

## In vivo protein interaction analysis

Cell extracts from asynchronously growing cultures were prepared in EBX buffer (50 mM HEPES-KOH pH 7.5, 100 mM KCl, 2.5 mM MgCl₂, 10% glycerol, 0.25% Triton X-100, 0.5 mM TCEP, protease inhibitors, RNase and benzonase) using glass beads breakage in a cooled Multi-Beads Shocker (Yasui Kikai). Extracts were cleared by centrifugation, precleared, and incubated with either IgG coated Dynabeads (ThermoFisher) for Protein A pulldown or with Protein A Dynabeads previously coated with anti-Pk antibody. Beads were washed three times with EBX buffer and once with EBX containing 300 mM KCl, then elution was carried out in SDS-PAGE loading buffer. Antibodies used for pulldown and immunoblotting were mouse monoclonal anti-HA clone F-7 (Santa Cruz Biotechnology sc-7392, used 1: 5,000), mouse monoclonal anti-HA HRP conjugated clone GG8-1F3.3.1 (Miltenyi Biotec 130-091-972, used 1: 2,000), mouse monoclonal anti-V5(Pk) clone SV5-Pk1 (BioRad MCA1360, used 1: 2,000), mouse monoclonal anti-AID antibody (2B Scientific CAC-APC004AM, used 1: 5,000) and rabbit polyclonal anti-Sth1 antibody (a kind gift from B. Cairns, used 1: 5,000). Following incubation with a peroxidase-coupled secondary antibody, blots were developed using enhanced chemiluminescence reagents (Cytiva) and visualized using an Amersham 600 imager and its control software (v.1.2.0).

## In vivo nucleosome positioning analysis

Mononucleosomal DNA isolation was performed as described[52]. Cells were fixed with formaldehyde, cell walls digested with Zymolase 100 T and unprotected DNA was digested with 30 U MNase for 20 min at 37 °C. DNA was purified, size separated by agarose gel electrophoresis and the band corresponding to mononucleosomal DNA was excised and processed for sequencing. Libraries were prepared using NEBNext Ultra II DNA Library Prep Kit for Illumina. 100 bp paired end sequencing of MNase-resistant DNA was performed on the Illumina HiSeq 4000 platform to generate ~20 million reads. Raw reads from each sample were adapter-trimmed using cutadapt (version 1.9.1)[53] with parameters -a: AGATCGGAAGAGC, -A: AGATCGGAAGAGC, minimum-length = 25, quality-cutoff = 20. BWA (version 0.5.9-r16)[54] with default parameters was used to perform genome-wide mapping of the adapter-trimmed reads to the yeast sacCer3 genome. Alignments were filtered to remove read pairs that were discordant, mapped to different chromosomes, ambiguously mapped, had an insert size outside the range 120–200 bp, or more than 4 mismatches in any read.

To generate heatmaps, genes were removed from each sample that did not achieve more than a + 0.6 nucleosome occupancy difference between the test sample and wild type. For the Degron vs. WT comparison in Supplementary Fig. 5a, theses were 29 genes. For the heatmaps in Fig. 4 the blacklists were 403, 634, 506 for Degron, +SCC2C and +SCC2N vs. WT respectively. The latter lists were merged without duplication, resulting in 1070 removed genes. The gene order in the heatmaps was then created by taking the mean difference of each gene between positions −150 and −50 for Degron vs. WT and ordering the genes by decreasing mean difference in this region. In case of Fig. 4, the ordering from Degron vs. WT was then applied to all heatmaps.

Sample-level smoothed coverage tracks for nucleosome profile plots were generated with the DANPOS2 dpos command (version 2.2.2)[55] with parameters paired: 1, span: 1, smooth width: 20, width: 40, count: 10,000,000. Either all genes, those genes reported to contain a promoter Scc2-Scc4 binding site[13], or the top 500 genes, were included in the analysis. The top 500 list contains genes with the highest mean nucleosome occupancy gain in the −150 and −50 region when comparing Degron vs. WT. In case of Fig. 4, a baseline of SCC2wt vs. WT was subtracted before ranking. The MNase, histone H4-ChIP data of in vivo formaldehyde-crosslinked cells[56] was used as the reference dataset for +1 nucleosome dyad locations.

## Reporting summary

Further information on research design is available in the Nature Portfolio Reporting Summary linked to this article.

## Data availability

The data that support this study are available from the corresponding authors upon reasonable request. The CLMS data generated in this study are contained in Supplementary Data 1, the raw data is available at the ProteomeXchange Consortium via the PRIDE partner repository, accession number PXD033446. The MNase sequencing datasets have been deposited with the Gene Expression Omnibus, accession number GSE197657. Source data are provided with this paper. The source data, which includes all unprocessed gel images and raw data has also been placed in the Mendeley repository where it can be accessed at https://data.mendeley.com/datasets/vhcrwy6z5f/draft?a=36d56553-aca0-4684-aabb-817a312eed9e.

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

## Acknowledgements

We would like to thank S. Mouilleron, F. Passarelli and M. Skehel for their help, A. Ballandras-Colas and P. Cherepanov, B. Cairns, C. Casas-Delucchi and J. Diffley, R. Dominguez and P. Carman, S. Hinshaw, and M. Minamino for reagents, the Crick Peptide Chemistry, Fermentation, Proteomics, and High Throughput Sequencing Science Technology Platforms, A. Bueno and M. Sacristán for their support and critical reading of the manuscript, as well as all our laboratory members for discussions. This work was supported by the European Research Council (ERC) under the Horizon 2020 program (grant agreement No. 670412), by The Francis Crick Institute (cc2137) receiving its core funding from Cancer Research UK, the UK Medical Research Council and the Wellcome Trust, as well as by a Programa II postdoctoral fellowship from the University of Salamanca and by the Spanish Ministry of Science grant PID2019-109616GB-I00 (S.M.).

## Author contributions

S.M. and F.U. conceived the study, S.M. performed all experiments, with help from C.B., A.J. performed the CLMS analyses, T.G. and H.P. analyzed the MNase-sequencing data, S.M. and F.U. wrote the manuscript with input from all coauthors.

## Funding

## Competing interests

The authors declare no competing interests.
