## [Peer Review File · Nature Communications]

REVIEWER COMMENTS

Reviewer #1 (Remarks to the Author):

Munoz et al. reported on cohesin loader as a regulator of chromatin remodeling. The authors showed that the cohesin loader Scc2-Scc4 promotes nucleosome remodeling activity by RSCs. As a mechanism for promoting this activity, they showed that Scc2-Scc4 binds to the ATPase domain of Sth1. The authors also demonstrated that Scc2-Scc4 can rescue the inhibition of nucleosome eviction in the promoter region caused by the depletion of the cohesin loader. Furthermore, the cohesin loader was found to interact with other chromatin remodelers (ISWI, SWI/SNF, CHD, and INO90), suggesting that the cohesin loader may regulate the chromatin landscape while interacting with chromatin remodelers. While the topic of chromatin remodeler regulation by cohesin loaders is of great interest, I feel that the biochemical and genomic analyses they provide are insufficient for publication in Nature Communication. In particular, the cohesin loader interacts with the RSC, but lacks convincing data to show that it is active. Extensive additional experiments would be required to confirm the authors' arguments, focusing on the following points.

Major points

1. Fig.3a and d do not appear to show an acceleration of nucleosome sliding by RSC1 with the addition of cohesin loader, and therefore, are not sufficient data to prove this. Also, the quantitative data and the intensity of the bands do not seem to match. Please clarify what criteria are used to show that there is a difference (e.g., the method of quantification) and it should be confirmed with different approaches such as adding ATP to an ATP null condition may be useful.
2. The effect of Promoter nucleosome eviction by the Cohesin loader seems to be limited. The amount of Sth1 actually enriched in the IP of the Fig2c loader is very low compared to Input. The authors need to validate this concern. For example, it is possible to extract by clustering the finely varying regions of Nucleosome occupancy in the Promoter region from the Degron and rescue experiments in Fig. 4b. By annotating the extracted regions, the significance of chromatin remodeler control by cohesin loader can be expected to be cleared up. Also, there is no biological replication and the replication should be increased to ensure reproducibility.
3. the authors lack experimental confirmation of the following statement. Further experimental confirmation should be performed to verify whether the FEDWF motif binds to the cohesin loader or whether changing the amino acids in the motif eliminates the binding. Homology analysis of the cohesin loader may also be useful. However, if this is only a possibility, it may be better to move the Figure to the Supplement rather than the Main. Finally, we studied sequence conservation of the cohesin loader-interacting peptides on the Sth1 ATPase domain surface. This revealed a highly conserved FEDWF motif in peptide 1 situated in the protrusion domain, both amongst chromatin remodeler families and between species (Fig. 5d). The motif includes a surface-exposed aromatic tryptophan, as well as surface- exposed negatively charged residues (Supplementary Fig. 5), opening the possibility that a conserved surface feature on The motif includes a surface-exposed aromatic tryptophan, as well as surface- exposed negatively charged residues (Supplementary Fig. 5), opening the possibility that a conserved surface feature on chromatin remodeler ATPases serves as a docking site for the cohesin loader.

Minor Points

1. please clarify the definition of color in Fig.1d.

Reviewer #2 (Remarks to the Author):

The paper by Muñoz and colleagues is well written and expands on the already known interaction between the RSC subunit Sth1 and the cohesin loader Scc2-Scc4. The authors characterize the domains important for this interaction and nicely show that the cohesin loader influences the chromatin remodeling activity of the RSC complex both in vivo and in vitro. However, few issues need to be addressed to strengthen the manuscript.

-Please specify the antibodies employed for immunoblotting in the method section

-Was the interaction between Scc2 and Sth1 picked up by CLMS? The interaction between the two subunits is confirmed by the authors, since they show that Scc2C alone can (albeit less efficiently) co-immunoprecipitate Sth1. Any speculation why this interaction is missed by CLMS? Is it possible that additional interactions between the RSC subunits and the cohesin loader subunits have been missed? Given the large size of the RSC complex, it would be expected that it contacts the cohesin loader at multiple sites

-What is the difference between the experiments performed in Figure 3a and in Supplementary Figure 3b? Do they contain different RSC paralogous subunits?

-The interaction between the cohesin loader and Sth1 as well as the stimulation of RSC chromatin remodeling activity seems to be mainly dependent on Scc4. However, the interaction with the different RSC1 or RSC2 complexes is assessed only for Scc2. Co-immunoprecipitation of Scc4 should be assessed as well

-Despite the major focus of the paper on Scc4, the authors performed fast conditional depletion of Scc2 instead of Scc4. It is known that the interaction between Scc2 and Scc4 is important for the stability of both proteins, and that previous studies have shown that depletion of one cohesin loader subunit by RNA interference leads to loss of the expression of the other subunit. However, the depletion of Scc4 in the degron cells should also be assessed and shown next to the Scc2 Immunoblot in Supplementary Figure 4b

-The authors nicely show that the cohesin loader complex is required for stimulation of the nucleosome remodeling activity of the RSC complex. Alterations of nucleosome positioning will inevitably affect cohesin loading and transcriptional regulation of several genes, a connection that is also explored by the authors in the paragraph of the discussion "The cohesin loader and gene regulation". However, the investigation of the consequences of depletion of RSC or cohesin loader subunits on transcription is missing in the manuscript. If the cohesin loader is required for stimulation of the RSC complex and nucleosome eviction in vivo, does the depletion of Sth1 lead to similar or partially overlapping consequences at the transcription level as the depletion of the cohesin loader?

Reviewer #3 (Remarks to the Author):

The cohesin complex plays crucial roles in genome organization, gene transcription, DNA repair, and sister chromatid cohesion. The cohesin loader, the Scc2-Scc4 complex, loads cohesin onto chromosomes at specific locations through interactions with chromatin receptors. The RSC chromatin remodeling complex has been identified as one such chromatin receptor for Scc2-Scc4. How the Scc2-Scc4 complex interacts with RSC remains poorly understood, however. In the current study, Muñoz et al. mapped the interactions between RSC and the cohesin loader Scc2-Scc4 and found a prominent contact between Scc4 and the conserved ATPase module of RSC. They then showed that Scc2-Scc4 stimulated the nucleosome remodeling activity of RSC in vitro and facilitated nucleosome

eviction at promoters in yeast cells. More interestingly, they identified a conserved motif in the chromatin-remodeling ATPases that might be important for binding to the cohesin loader.

Overall, this study provides good evidence to suggest a new function of the cohesin loader, i.e. the stimulation of chromatin remodeling ATPases. This finding has far-reaching implications and should be published in a major journal. Certain results are not terribly convincing, however. The following specific points need to be addressed prior to publication.

Specific points

- 1. Only a single crosslink between Scc4 and Sth1 was detected by crosslinking mass spectrometry (CLMS). This is not very convincing. The authors later showed that the interaction between RSC and Scc2-Scc4 is strengthened in the presence of ADP · BeF3. They should repeat the CLMS study with the addition of ADP · BeF3.**
- 2. The authors claim that the cohesin loader stimulates the chromatin remodeling activities of RSC1/2 (Figure 3a and 3d). The effects were barely noticeable from the gel images. The quantification appeared to reveal a larger effect. The authors should describe their quantification methods. They should optimize the assay to obtain more compelling results.**
- 3. In Figure 4b, the differences between the profiles are very small. Are these small differences reproducible?**
- 4. The authors have identified a highly conserved FEDWF motif in chromatin-remodeling ATPases. They should test whether mutations of this motif in RSC1/2 disrupt Scc2-Scc4 binding.**
- 5. In the discussion, the authors suggest that the CdLS mutations might cause the disease phenotypes by affecting chromatin remodeling, as opposed to cohesin loading. They should be more cautious in making this claim. Although NIPBL mutations are the most frequent among CdLS patients, mutations in other cohesin subunits are also linked to CdLS. It is very likely that a partial loss of cohesin function underlies CdLS and other cohesinopathies.**

We would like to thank the three reviewers for their interest in our study, as well as for their constructive critique. In response, we have expanded our investigations, including an additional protein crosslink mass spectrometry experiment, a comprehensive mutational analysis of the FEDWF remodeler interaction motif, further biochemical experiments, as well as analyses of the micrococcal nuclease sequencing data. All our additional experiments and analyses are detailed in the below point-by-point response.

(As per the editor's request, we include *the full text of the reviews in blue italic*, to which our responses are added in black)

Reviewer 1

Munoz et al. reported on cohesin loader as a regulator of chromatin remodeling. The authors showed that the cohesin loader Scc2-Scc4 promotes nucleosome remodeling activity by RSCs. As a mechanism for promoting this activity, they showed that Scc2-Scc4 binds to the ATPase domain of Sth1. The authors also demonstrated that Scc2-Scc4 can rescue the inhibition of nucleosome eviction in the promoter region caused by the depletion of the cohesin loader. Furthermore, the cohesin loader was found to interact with other chromatin remodelers (ISWI, SWI/SNF, CHD, and INO90), suggesting that the cohesin loader may regulate the chromatin landscape while interacting with chromatin remodelers. While the topic of chromatin remodeler regulation by cohesin loaders is of great interest, I feel that the biochemical and genomic analyses they provide are insufficient for publication in Nature Communication. In particular, the cohesin loader interacts with the RSC, but lacks convincing data to show that it is active. Extensive additional experiments would be required to confirm the authors' arguments, focusing on the following points.

Major points

1. Fig.3a and d do not appear to show an acceleration of nucleosome sliding by RSC1 with the addition of cohesin loader, and therefore, are not sufficient data to prove this. Also, the quantitative data and the intensity of the bands do not seem to match. Please clarify what criteria are used to show that there is a difference (e.g., the method of quantification) and it should be confirmed with different approaches such as adding ATP to an ATP null condition may be useful.

To address the reviewer's concern about quantification of the *in vitro* nucleosome remodeling assay, we have added a new panel (Supplementary Fig. 3c) to illustrate this analysis. The panel clarifies the identities of the input and product bands, as well as their quantification. As suggested by the reviewer, we have also performed a reaction in the absence of ATP, to further confirm the specificity of this biochemical assay (Supplementary Fig. 3b). To facilitate further inspection, we have posted original gel images of all repeat experiments with the Mendeley repository.

2. The effect of Promoter nucleosome eviction by the Cohesin loader seems to be limited. The amount of Sth1 actually enriched in the IP of the Fig2c loader is very low compared to Input. The authors need to validate this concern. For example, it is possible to extract by clustering the finely varying regions of Nucleosome occupancy in the Promoter region from the Degron and rescue experiments in Fig. 4b. By annotating the extracted regions, the significance of chromatin remodeler control by cohesin loader can be expected to be cleared up. Also, there is no biological replication and the replication should be increased to ensure reproducibility.

This point covers two aspects of our analysis. Firstly, the reviewer finds that the interaction between Sth1 and the cohesin loader, shown in Fig. 2c, is weak. We would like to emphasize that the interaction analysis uses Coomassie Blue staining to visualize both bait and prey on the same gel. The amount of coprecipitation must be substantial to be detectable in this way. We estimate that around 10 – 20% of the cohesin loader copurifies with Sth1, which constitutes a very good yield for two proteins that engage in a transient interaction.

A second concern relates to the micrococcal nuclease sequencing experiment shown in Figure 4. Promoter nucleosome occupancy was only weakly affected following cohesin loader depletion. The small effect size stemmed from the averaging across all genes, which diluted the impact. We have therefore reanalyzed the data and now show averages of the 500 most affected genes in Fig. 4b (averages of all genes, and of genes previously reported as cohesin loader binding sites, can be found in Supplementary Fig. 5b). This depiction allows a better comparison between the rescue effects of the cohesin loader modules (Scc2C or Scc2N-Scc4, or the full cohesin loader). Note that the impact of the cohesin loader on promoter nucleosome positioning is reproducible and in confirmation of previously published results (Lopez-Serra et al. 2014, Nat. Genet. 46, 1147). The main purpose of the current experiment is to compare the respective contributions of the two cohesin loader modules. The results show that both modules contribute to shaping the *in vivo* promoter chromatin landscape.

3. The authors lack experimental confirmation of the following statement. Further experimental confirmation should be performed to verify whether the FEDWF motif binds to the cohesin loader or whether changing the amino acids in the motif eliminates the binding. Homology analysis of the cohesin loader may also be useful. However, if this is only a possibility, it may be better to move the Figure to the Supplement rather than the Main. Finally, we studied sequence conservation of the cohesin loader-interacting peptides on the Sth1 ATPase domain surface. This revealed a highly conserved FEDWF motif in peptide 1 situated in the protrusion domain, both amongst chromatin remodeler families and between species (Fig. 5d). The motif includes a surface-exposed aromatic tryptophan, as well as surface-exposed negatively charged residues (Supplementary Fig. 5), opening the possibility that a conserved surface feature on the motif includes a surface-exposed aromatic tryptophan, as well as surface-exposed negatively charged residues (Supplementary Fig. 5), opening the possibility that a conserved surface feature on chromatin remodeler ATPases serves as a docking site for the cohesin loader.

The reviewer raises a key point when asking for an experimental investigation whether the FEDWF motif indeed mediates the cohesin loader – Sth1 interaction. To address this question, we have generated a mutational peptide array in which each amino acid in the region surrounding the FEDWF motif was mutated into one of four alternative amino acids. This analysis revealed that the FEDWF motif (together with three additional downstream amino acids) is indeed responsible for the cohesin loader interaction (see the new Supplementary Fig. S6b).

We also placed FEDWF motif mutations into the context of the intact RSC complex. However, as expected from the multipronged interactions between RSC and the cohesin loader, a residual interaction remained detectable even following FEDWF motif mutation.

Despite this, the FEDWF motif is crucial for RSC function and regulation (Clapier et al. 2020, Mol. Cell 80, 712), placing Scc4 at a crucial point to take part in this regulation.

As further suggested by the reviewer, we characterized the putative Sth1 interaction site on Scc4. A sequence alignment reveals some conservation in this otherwise very poorly conserved cohesin loader subunit. Furthermore, mutational peptide scans revealed the importance of i) a motif that is also involved in the Scc4 interaction with its Ctf19 centromere receptor, and ii) a previously unannotated, conserved KLIK motif (see the new Supplementary Fig. S6c).

Minor Points

1. please clarify the definition of color in Fig.1d.

Apologies for the inconsistent colors used in the Fig. 1d labels, which have been corrected.

Reviewer 2

The paper by Muñoz and colleagues is well written and expands on the already known interaction between the RSC subunit Sth1 and the cohesin loader Scc2-Scc4. The authors characterize the domains important for this interaction and nicely show that the cohesin loader influences the chromatin remodeling activity of the RSC complex both in vivo and in vitro. However, few issues need to be addressed to strengthen the manuscript.

-Please specify the antibodies employed for immunoblotting in the method section

Details of all the antibodies used in our study can now be found in the Methods section (in addition to the Reporting Summary).

-Was the interaction between Scc2 and Sth1 picked up by CLMS? The interaction between the two subunits is confirmed by the authors, since they show that Scc2C alone can (albeit less efficiently) co-immunoprecipitate Sth1. Any speculation why this interaction is missed by CLMS? Is it possible that additional interactions between the RSC subunits and the cohesin loader subunits have been missed? Given the large size of the RSC complex, it would be expected that it contacts the cohesin loader at multiple sites

The reviewer raises a fair point, when suggesting that interactions between RSC and the Scc2C module of the cohesin loader should also be detectable by crosslink mass spectrometry (CLMS). We have repeated the CLMS analysis which has indeed uncovered an additional interaction of RSC (the N-terminus of the Sth1 ATPase) with a surface patch on the Scc2 hook. These results have been incorporated into the revised Figs. 1b and c.

-What is the difference between the experiments performed in Figure 3a and in Supplementary Figure 3b? Do they contain different RSC paralogous subunits?

The experiments shown in Fig. 3a and Supplementary Fig. 3b were essentially the same. The latter was included to illustrate reproducibility. During the revisions (and in response to points raised by reviewers 1 and 3), we have repurposed Supplementary Fig. 3b (now 3c) to illustrate the quantification used to compare chromatin remodeling in the absence and presence of the cohesin loader.

-The interaction between the cohesin loader and Sth1 as well as the stimulation of RSC chromatin remodeling activity seems to be mainly dependent on Scc4. However, the interaction with the different RSC1 or RSC2 complexes is assessed only for Scc2. Co-immunoprecipitation of Scc4 should be assessed as well

We agree with the reviewer that our data suggest that both Scc4 and Scc2 make contact with RSC (substantiated by the additional crosslinking mass spectrometry data in the updated Fig. 1). As suggested by the reviewer, we have also repeated our interaction experiments to monitor Scc4 coimmunoprecipitation with RSC1 and RSC2. Unfortunately, the electrophoretic mobility of Scc4 overlaps with the immunoglobulin heavy chain, and we were unable to discern Scc4 in this experiment. We should stress, however, that Scc2 and Scc4 are always observed in a tight complex, as expected from their crystallographically observed, intertwined nature.

-Despite the major focus of the paper on Scc4, the authors performed fast conditional depletion of Scc2 instead of Scc4. It is known that the interaction between Scc2 and Scc4 is important for the stability of both proteins, and that previous studies have shown that depletion of one cohesin loader subunit by RNA interference leads to loss of the expression of the other subunit. However, the depletion of Scc4 in the degron cells should also be assessed and shown next to the Scc2 Immunoblot in Supplementary Figure 4b

As suggested by the reviewer, we monitored Scc4 levels during acute, auxin-induced Scc2 depletion. To our surprise, Scc4 levels remained unaltered during the observation period (see the updated Supplementary Fig. 5). This finding suggests that Scc4 is proteolytically cleared from cells only at longer times without Scc2. However, despite persistence of the polypeptide, it is unlikely that Scc4 retains a functional fold in the absence of Scc2 (Chao et al. 2015, Cell Rep. 12, 719; Hinshaw et al. 2015, Elife 4, e06057).

-The authors nicely show that the cohesin loader complex is required for stimulation of the nucleosome remodeling activity of the RSC complex. Alterations of nucleosome positioning will inevitably affect cohesin loading and transcriptional regulation of several genes, a connection that is also explored by the authors in the paragraph of the discussion "The cohesin loader and gene regulation". However, the investigation of the consequences of depletion of RSC or cohesin loader subunits on transcription is missing in the manuscript. If the cohesin loader is required for stimulation of the RSC complex and nucleosome eviction in vivo, does the depletion of Sth1 lead to similar or partially overlapping consequences at the transcription level as the depletion of the cohesin loader?

The reviewer raises a very pertinent point. If the cohesin loader acts by promoting RSC chromatin remodeling, one should expect related transcriptional changes following cohesin loader or RSC depletion. This is indeed the case, the transcriptional changes in following cohesin loader or RSC inactivation are strongly correlated in budding yeast (Lopez-Serra et al. 2014, Nat. Genet. 46, 1147). Even human patient etiologies in response to cohesin loader (Cornelia de Lange Syndrome) or RSC ortholog mutations (Coffin-Siris Syndrome) are strikingly overlapping. This is more clearly laid out in the revised discussion.

Reviewer 3

The cohesin complex plays crucial roles in genome organization, gene transcription, DNA repair, and sister chromatid cohesion. The cohesin loader, the Scc2-Scc4 complex, loads

cohesin onto chromosomes at specific locations through interactions with chromatin receptors. The RSC chromatin remodeling complex has been identified as one such chromatin receptor for Scc2-Scc4. How the Scc2-Scc4 complex interacts with RSC remains poorly understood, however. In the current study, Muñoz et al. mapped the interactions between RSC and the cohesin loader Scc2-Scc4 and found a prominent contact between Scc4 and the conserved ATPase module of RSC. They then showed that Scc2-Scc4 stimulated the nucleosome remodeling activity of RSC in vitro and facilitated nucleosome eviction at promoters in yeast cells. More interestingly, they identified a conserved motif in the chromatin-remodeling ATPases that might be important for binding to the cohesin loader.

Overall, this study provides good evidence to suggest a new function of the cohesin loader, i.e. the stimulation of chromatin remodeling ATPases. This finding has far-reaching implications and should be published in a major journal. Certain results are not terribly convincing, however. The following specific points need to be addressed prior to publication.

Specific points

1. Only a single crosslink between Scc4 and Sth1 was detected by crosslinking mass spectrometry (CLMS). This is not very convincing. The authors later showed that the interaction between RSC and Scc2-Scc4 is strengthened in the presence of ADP · BeF₃. They should repeat the CLMS study with the addition of ADP · BeF₃.

In response to the reviewer's suggestion, we have replicated the CLMS analysis. The repeat analysis has uncovered an additional interaction of RSC (the N-terminus of the Sth1 ATPase) with a surface patch on the Scc2 hook. These results are consistent with the notion that both Scc4 and Scc2 contribute to the RSC interaction and have been incorporated into the revised Figs. 1b and c.

We have also compared CLMS data obtained in the presence and absence of ADP·BeF₃, however this did not lead to notable differences in the crosslink spectrum.

2. The authors claim that the cohesin loader stimulates the chromatin remodeling activities of RSC1/2 (Figure 3a and 3d). The effects were barely noticeable from the gel images. The quantification appeared to reveal a larger effect. The authors should describe their quantification methods. They should optimize the assay to obtain more compelling results.

To address the reviewer's concern about quantification of the *in vitro* nucleosome remodeling assay, we have added a new panel (Supplementary Fig. 3c) to illustrate this analysis. The panel clarifies the identities of the input and product bands, as well as their quantification. As suggested by the reviewer, we have also performed a reaction in the absence of ATP, to further confirm the specificity of this biochemical assay (Supplementary Fig. 3b). To facilitate further inspection, we have posted original gel images of all repeat experiments with the Mendeley repository.

3. In Figure 4b, the differences between the profiles are very small. Are these small differences reproducible?

This concern relates to the micrococcal nuclease sequencing experiment, in which promoter nucleosome occupancy was only weakly affected following cohesin loader depletion. The small effect size stemmed from the averaging across all genes, which diluted the impact. We have therefore reanalyzed the data and now show averages of the 500 most affected genes

in Fig. 4b (averages of all genes, and of genes previously reported as cohesin loader binding sites, can be found in Supplementary Fig. 5b). This depiction allows a better comparison between the rescue effects of the cohesin loader modules (Scc2C or Scc2N-Scc4, or the full cohesin loader). Note that the impact of the cohesin loader on promoter nucleosome positioning is reproducible and in confirmation of previously published results (Lopez-Serra et al. 2014, Nat. Genet. 46, 1147). The main purpose of the current experiment is to compare the respective contributions of the two cohesin loader modules. The results show that both modules contribute to shaping the *in vivo* promoter chromatin landscape.

4. The authors have identified a highly conserved FEDWF motif in chromatin-remodeling ATPases. They should test whether mutations of this motif in RSC1/2 disrupt Scc2-Scc4 binding.

The reviewer raises a key point when asking for an experimental investigation whether the FEDWF motif indeed mediates the cohesin loader – Sth1 interaction. To address this question, we have generated a mutational peptide array in which each amino acid in the region surrounding the FEDWF motif was mutated into one of four alternative amino acids. This analysis revealed that the FEDWF motif (together with three additional downstream amino acids) is indeed responsible for the interaction (see the new Supplementary Fig. S6b).

We also placed FEDWF motif mutations into the context of the intact RSC complex. However, as expected from the multipronged interactions between RSC and the cohesin loader, a residual interaction remained detectable even following FEDWF motif mutation. Despite this, the FEDWF motif is crucial for RSC function and regulation (Clapier et al. 2020, Mol. Cell 80, 712), placing Scc4 at a crucial point to take part in this regulation.

5. In the discussion, the authors suggest that the CdLS mutations might cause the disease phenotypes by affecting chromatin remodeling, as opposed to cohesin loading. They should be more cautious in making this claim. Although NIPBL mutations are the most frequent among CdLS patients, mutations in other cohesin subunits are also linked to CdLS. It is very likely that a partial loss of cohesin function underlies CdLS and other cohesinopathies.

We appreciate the reviewer's caution when it comes to the important question how cohesin and cohesin loader mutations cause cohesinopathies. We agree with the reviewer that a key contribution no doubt stems from compromised cohesin function. This is now clearly stated in our discussion. Others have proposed involvement of, e.g., the super elongation complex (Izumi et al. 2015, Nat. Genet. 47, 338). We would like to propose that the nucleosome landscape might also make a contribution to the transcriptional changes observed in Cornelia de Lange syndrome (note that the cohesin complex itself interacts with RSC, which could underpin effects of cohesin patient mutations). Our revised discussion better presents these, not mutually exclusive, possibilities.

REVIEWER COMMENTS

Reviewer #1 (Remarks to the Author):

authors' claim that cohesin loaders interact with chromatin remodelers and activate remodeling is not yet well presented for reproducibility.

1. In the nucleosome remodeling experiment (Fig. 3), the authors' newly added Supplemental Figure 3c is important for understanding the quantitative method. It is appropriate to add it to the main figure. In addition, to clarify the reproducibility of the experiment, it is recommended that all gel images of replicates, especially Fig. 3a & d, be included in the Supplemental Figure. 2.

2. for Fig.4, more than duplicate data acquisition should be performed to verify the reproducibility of the data.

3. Regarding the claim that the cohesin loader binds to the ATPase domain of RSC (Fig. 2 c), it is confirmed by IP that the cohesin loader binds to the Sth1365-1097 fragment, but in order to claim "efficiently recovered" in 161-162 of the text, the rationale for the "efficiently" should be explained in the text. However, it would be better to tone down the wording a little if there is no basis for the claim.

Reviewer #2 (Remarks to the Author):

I am satisfied with most of the authors' responses to my concerns.

However, I would like to stress one point. In response to my request to monitor Scc4 levels in the degran cells, the authors interestingly show that Scc4 is still expressed/could still be detected up to two hours after acute depletion of Scc2. This is indeed surprising. It would be interesting to monitor Scc4 expression for a longer period of time after acute Scc2 depletion to verify whether (and when) degradation of Scc4 occurs. In addition, the authors speculate that Scc4 of the degran cells is unlikely to retain a functional fold in the absence of Scc2 (and would therefore be expected to be non-functional). This statement should be verified/demonstrated. One possibility would be to test the interaction of Sth1 with Scc4 in the degran cells through coimmunoprecipitation.

In this regard (and also in response to the authors' comments stating that "the electrophoretic mobility of Scc4 overlaps with the immunoglobulin heavy chain, and we were unable to discern Scc4 in this experiment"), I would like to suggest that there are secondary antibodies available (for example from Abcam) that are specific for IPs and allow to overcome this issue by detecting only the native form of the primary antibodies without interference from denatured IgG.

Reviewer #3 (Remarks to the Author):

The authors have adequately addressed my concerns. Publication is recommended.

We would like to thank the three reviewers for their feedback on our revised manuscript. We were happy to see their overall satisfaction with our revisions. The first two reviewers included suggestions for further improvements to our study. In response, we have carried out a second round of revisions. Please find below a point-by-point response how we have addressed each of the remaining concerns.

Reviewer #1 (Remarks to the Author):

authors' claim that cohesin loaders interact with chromatin remodelers and activate remodeling is not yet well presented for reproducibility.

1. In the nucleosome remodeling experiment (Fig. 3), the authors' newly added Supplemental Figure 3c is important for understanding the quantitative method. It is appropriate to add it to the main figure. In addition, to clarify the reproducibility of the experiment, it is recommended that all gel images of replicates, especially Fig. 3a & d, be included in the Supplemental Figure. 2.

We agree with the reviewer that the quantitative method used in Figure 3 could be even better explained. For this purpose, as the reviewer suggests, we have moved Supplementary Figure 3c to the main figures, where it has become the new Figure 3a.

To clarify the reproducibility of the experiment, we agree that the gel images of all replicates must be available. All the gel images were deposited and are available from the Mendeley repository. A weblink is provided in the manuscript. Alternatively, we are of course happy to include the full set of gel images as part of the Supplementary Figure 3, provided the editor agrees that this would not unduly increase the size of this Supplementary Figure. Please advise.

2. for Fig.4, more than duplicate data acquisition should be performed to verify the reproducibility of the data.

The reviewer asks for an additional repeat of the micrococcal nuclease sequencing experiment, confirming that the cohesin loader indeed contributes to maintaining nucleosome-free gene promoters. In response, we have performed a repeat experiment, which has confirmed reproducibility of the results contained in our manuscript (as well as previously published by Lopez-Serra et al. 2014 *Nat. Genet.* **46**, 1147-1151). The results from this additional micrococcal nuclease sequencing experiment are shown in the new Supplementary Figure 6. The primary data has been added to our GEO submission.

3. Regarding the claim that the cohesin loader binds to the ATPase domain of RSC (Fig. 2 c), it is confirmed by IP that the cohesin loader binds to the Sth1365-1097 fragment, but in order to claim "efficiently recovered" in 161-162 of the text, the rationale for the "efficiently" should be explained in the text. However, it would be better to tone down the wording a little if there is no basis for the claim.

We agree with the reviewer that it is hard to quantify interaction 'efficiency' based on band intensities in a semi-quantitative pull-down experiment. We have therefore omitted the word 'efficiently' in our revised manuscript.

Reviewer #2 (Remarks to the Author):

I am satisfied with most of the authors' responses to my concerns.

However, I would like to stress one point. In response to my request to monitor Scc4 levels in the degron cells, the authors interestingly show that Scc4 is still expressed/could still be detected up to two hours after acute depletion of Scc2. This is indeed surprising. It would be interesting to monitor Scc4 expression for a longer period of time after acute Scc2 depletion to verify whether (and when) degradation of Scc4 occurs. In addition, the authors speculate that Scc4 of the degron cells is unlikely to retain a functional fold in the absence of Scc2 (and would therefore be expected to be non-functional). This statement should be verified/demonstrated. One possibility would be to test the interaction of Sth1 with Scc4 in the degron cells through coimmunoprecipitation.

We would like to thank the reviewer for pointing out this omission. Like the reviewer, we were surprised that Scc4 remained stable following Scc2 depletion. Our depletion experiments (including the example shown in Supplementary Figure S5) were performed in G1 arrested cells. If a similar depletion experiment is performed while cells transition from G1 through S phase and into mitosis, Scc4 becomes unstable, and its levels decrease. This is documented in Munoz et al. 2019, *Mol. Cell* **74**, 664–673 (Figures 5B and D there). The same publication shows that Scc4 on its own does not load cohesin onto chromosomes nor promote sister chromatid cohesion (Supplementary Figures S5D and E there). Importantly, and in response to the reviewer's specific request for clarification, Scc4 on its own does not interact with Sth1 (see Figure 5D there). We agree that these are important pieces of background information, required to evaluate the nucleosome positioning experiment. We provide this information in the revised Supplementary Figure S5 legend.

In this regard (and also in response to the authors' comments stating that "the electrophoretic mobility of Scc4 overlaps with the immunoglobulin heavy chain, and we were unable to discern Scc4 in this experiment"), I would like to suggest that there are secondary antibodies available (for example from Abcam) that are specific for IPs and allow to overcome this issue by detecting only the native form of the primary antibodies without interference from denatured IgG.

We appreciate the reviewer's interest in an additional protein interaction experiment to probe the interaction of RSC1 and RSC2 with the cohesin loader. The manuscript contained data demonstrating an interaction of both RSC1 and RSC2 with the cohesin loader by detecting its Scc2 subunit (Figure 3c). We have during the second round of revisions generated an additional set of yeast strains with Scc4 epitope tags. This has allowed us to additionally document the interaction of RSC1 and RSC2 with the cohesin loader by detecting its Scc4 subunit. This was possible by using an anti-HA epitope antibody directly coupled to horseradish peroxidase. The additional interaction analysis is contained in a new Supplementary Figure S4b. Antibody information is included in the Methods, as well as in the Reporting Summary.

Reviewer #3 (Remarks to the Author):

The authors have adequately addressed my concerns. Publication is recommended.